# Use It Sustainably or Lose It! The Land Stakes in SDGs for Sub-Saharan Africa

**Cheikh Mbow [1,2]**

[1]  Future Africa, University of Pretoria, South St, Koedoespoort 456-Jr, Pretoria 0186, South Africa; cmbow@start.org; Tel.: +12026441101

[2]  College of Agriculture and Natural Resources, Department of Forestry, Michigan State University, Natural Resources Building 480 Wilson Road, Room 126, East Lansing, MI 48824, USA

**Abstract:** Sub-Saharan Africa (SSA) failed to meet most Millennium Development Goals (MDGs). The Sustainable Development Goals (SDGs) require knowledge-intensive actions that weigh development goals against sustainability options with several possibilities in various contexts. Land resources are the mainstay for most African communities and the basis of achievement of most SDGs. The "transformation imperative" in Africa will only take place in a differentiated set of resource management and use. The baselines in African countries are rather low in terms of internal policy and economic functions. The objective of this paper is to instate ideas on ways to achieve the SDGs through a new transformative design based on a collective capacity of diverse actors to access a range of land-based practices. We should selectively adapt, adopt, or consolidate various land innovations by targeting place and time where various practices have worked or can work in a range of ecologies; what seems to work over the short-term but reduces risks for the long-term; and what the implications are for wealth, food production, livelihoods, climate change, resilience, and development. This requires a greater capacity to apply what is known about transformative action but also set a collaborative learning system to influence policy-makers and action-takers to support sustainable transformation.

**Keywords:** land resources; transformation; food security; sustainable development; Sub-Saharan Africa

## 1. Introduction

Sub-Saharan Africa (SSA) failed to meet most Millennium Development Goals (MDGs) [1]. The reasons for the failure of the Millennium Development Goals (MDGs) are still prevalent in Sub-Saharan Africa (SSA) [1,2]. The weakness of the means of implementation following the "whatever gets you there" kind of scenario constitutes a big challenge [3], but there are many opportunities to innovate quicker, adopt faster, and deeply transform the continent if some of the many functions of land are harnessed for sustainable solutions [4,5]. With the Sustainable Development Goals (SDGs), SSA is committed to "end hunger, achieve food security and improved nutrition and promote sustainable agriculture", "protect, restore and promote sustainable use of terrestrial ecosystems, sustainably manage forests, combat desertification, and halt and reverse land degradation and halt biodiversity loss", "ensure availability and sustainable management of water", and at the same time "take urgent action to combat climate change and its impacts" [6]. All these challenges depend to a very large extent on the capacity to embrace a deep transformation to accelerate land performances. There are 17 SDGs with 232 indicators (revised number); targeting which one is most needed in various contexts and which indicator requires more attention will help avoid the dilution factor and the evasive implementation of the SDGs.

Dependency on land resources is one of the unwavering characteristics of Sub-Saharan Africa. Land is a place where the SDGs' ambitions could be addressed together and simultaneously. Achieving a positive future in Africa depends largely on land resources and requires sustainability knowledge on land resources management based on livelihood systems to support social and natural assets [7,8]. The effort and resources to reach SDGs will be a daunting endeavor because of the low baseline of African countries. Index and Dashboards synthesizing the starting point show a huge gap in Africa. While countries such as South Africa have the highest baseline score of 53.8 (out of a max of 100), others countries such as Nigeria (36.1), Chad (31.8), or Central African Republic (26.1) show the lowest baseline scores of the world [9].

At the same time, the time window to bridge the gap is rather short, only 10 years from when this paper was written. This means only 10 growing seasons to double food production where lands are largely arid or semi-arid and almost all parts are very vulnerable to climate change, land degradation, and deforestation [10,11]. Therefore, compared to other baselines from other continents, Africa has the steepest slope and the largest array of challenges to face. Africa also has the greatest opportunity to make the leap directly to prosperity without severe environmental and social degradation [12,13] because of its exceptional richness of land resources [8].

Deep and rapid transformation of Africa could benefit from a set of already known innovations such as land degradation neutrality, improved land management and agroforestry, and sustainable intensification [14] that can support the process of optimal use of land. Sustainable use of land involves increasing the productivity of agriculture while minimizing any negative economic, social, or environmental consequences [13]. The critical component of this pathway is the utilization of the existing land to produce greater yields, better nutrition, and higher net incomes while reducing over reliance on pesticides and fertilizers and lowering emissions of harmful greenhouse gases [15].

The profound transformation needed in Africa should be based on a new type of collective capacity supporting the African Regional Strategies on agriculture and food security (Comprehensive African Agriculture Development Program (CAADP)) [16] and the African Union Vision 2063 [17]. According to these strategic goals, food production in SSA must double in the coming decades to feed the continent, serve as an engine of growth, and adapt to climate change [18]. Extreme poverty eradication and food security in Africa are strongly linked to land productivity under various drivers of land dynamics. Improving land resource management is therefore at the heart of transformational change to pursue increased products delivery and services from land for poor communities [18]. The continent could achieve biosphere-positive development through food security and adaptation to climate change. Improved land management will improve livelihoods and sustain a vibrant economy while preserving ecosystem and biodiversity integrity [19]—but can this be done in just a decade?

Currently, Africa's rapidly developing academic and research base supports the SDGs and drives sustainable economic development, but not without a level of systemic thinking that only exists in theory and needs to be brought into daily development practices [20,21]. The deep and rapid social-environmental transformation—a great acceleration of development based on sustainable approaches—is not only a techno-optimistic stand in regards to advantages of the Fourth Industrial Revolution (4IR) [22], but a vivid dynamic driven by massive adoption of ways land is used to address various development options. The issue is how to positively influence the current trajectory of Africa and harness expected targets in various development areas.

Land is the element that determines whether SDGs are achieved in Africa, as land resources are the center of most development policies. SDGs to eradicate extreme poverty and hunger and address climate change and biodiversity, are strongly linked to land productivity risked by land degradation [1]. This paper suggests some avenues for rethinking the current use of land for food, energy, income-based products, and land use change including urbanization processes. The aim of this paper is to set the tone for a new rhetoric challenging the status quo in order to advance the implementation gap between land resources and land restoration [23] that helps accelerate the achievement of SDGs from land resources.

This paper addresses how Africa could develop and consolidate winning land practices to address poverty, growing population, and rapid urbanization. The main endeavor is how to promote implementation capacities for various innovations and economic opportunities that are coupled to Africa's knowledge system in a way that drives radical transformation towards sustainable prosperity.

There are three points of analysis in this paper: The first shows where we are in terms of facts related to food and poverty as a way to set the baseline against which we should assess the expected effort for achieving SDGs (Sections 1 and 2). The second revisits the topics of current knowledge, namely demographic growth, associated rapid urbanization, and response to climate change (Section 3). The third embraces various opportunities for land-based transformation that are relevant to SDGs (Sections 4 and 5). The conclusion discusses some proposals for moving forward.

## 2. Poverty and Food Security

Africa's baseline, against which SDGs are assessed, should be considered an invite to engage in a massive transformation to achieve a positive future. The task is to minimize the gap between Africa's current development trajectory, and the aims of the SDGs that consider climate and environment in the new development model. Understanding the magnitude of the gap between current trends and desired outcomes will help alleviate the tension between sustainability and the call for establishing a strong economy (Figure 1) [24]. The bottom-line is that development in Africa is mostly based on the extraction of land-based resources, in particular mines, forest, and agriculture, to address poverty, rapid demographic growth, and urbanization [1,25]. The development of Africa, shown in few selected examples, will help in the planning of land-based interventions for achieving various SDGs.

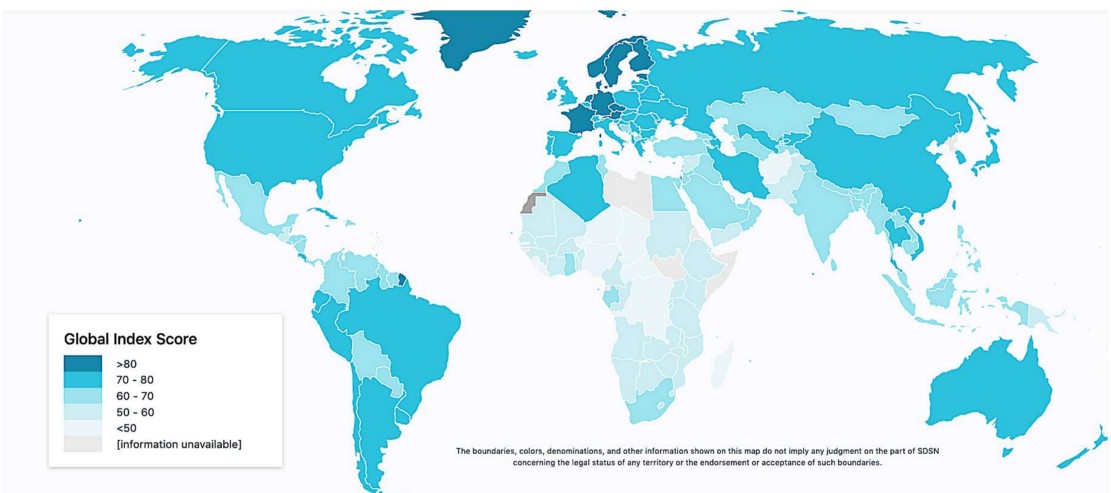

**Figure 1.** Global Index Report (African countries have the lowest scores, below 50% in most of them). Source: https://dashboards.sdgindex.org/#/ [26]

Poverty in Africa is persistent. Currently, about 48% of Africa's population, or approximately 450 million people, live in extreme poverty (less than US $1.25 per day), with 63% of the continent's poor living in rural areas depending on agriculture for their livelihoods [27]. About 75% of the world's poorest countries are in Africa. The ten countries with the highest proportion of people living in extreme poverty are in Sub-Saharan Africa [28]. About 70% of African countries (40 of them) have more than a 40% poverty rate showing that a large majority of the country currently has a severe poverty rate with a rural–urban divide (Figure 2). Most of the poor dwellers in SSA rely on annual crops and ecosystem services for their survival. The rural poor poverty rate is, in general, 2 to 6-times more than the urban poverty rate. In this context, rural migration into cities becomes a mere transfer of poverty with additional social consequences.

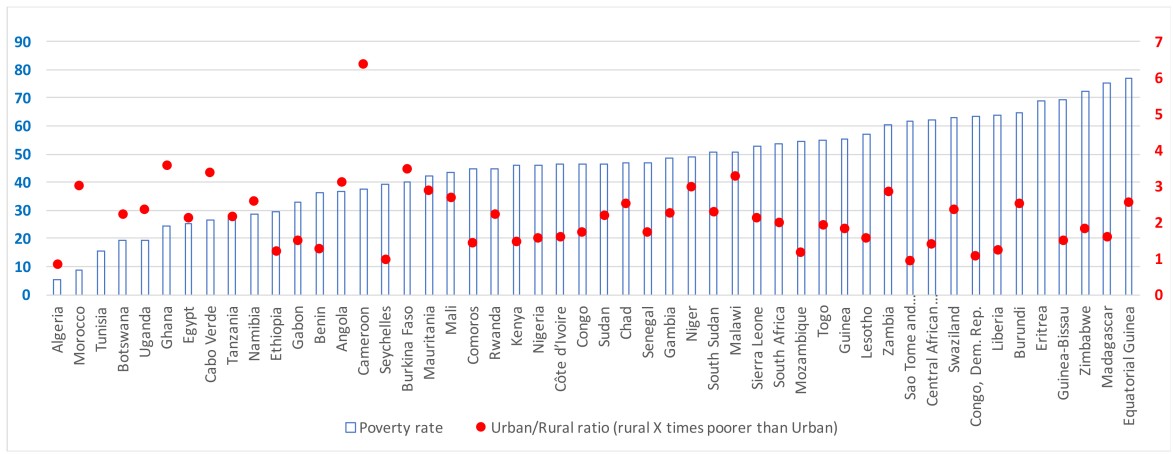

**Figure 2.** Poverty rate and urban/rural poverty difference (Source: [28].

This poverty baseline happens in a context where most countries rely on agriculture and forest resources to support poor dwellers. Africa's poor use land-based resources to generate most of their income. Land management to improve lives and livelihoods includes improved forest, farming, wetlands, and grazing systems. Most of the land management activities address issues of ecosystem services, soil erosion, and preservation of biodiversity. Land offers a huge opportunity to identify and enable the adoption and implementation of productive, equitable, and sustainable land management (SLM) for poverty reduction. Deforestation for agriculture is one of the strongest drivers for land use change [29–32] (Figure 5) because of low agricultural inputs, extensive pastoral systems, and rapid expansion of human settlements. The increase in income and food will only be significant if land-based products are sustainably used.

The impact of climate change accentuates poverty and population vulnerability to climate and non-climatic shocks. It is estimated, given the current warming trends in Sub-Saharan Africa, that the production of major cereals could decline by as much as 20% by mid-century [11]. The poor who depend on agriculture for their livelihoods and are less able to adapt will be disproportionately affected.

Food demand is known to be a major problem in Africa where over 20% (237 million people in Sub-Saharan Africa) were undernourished in 2017 [33]. According to the Food and Agriculture Organization (FAO) [33], hunger in Africa continues to rise, threatening the continent's hunger eradication efforts to meet SDG 2. There are many barriers to food security. First, the lack of a food system approach limits the attention given to trade and transport, how agriculture inputs are produced and used, access water and transformation food products, and consumption patterns [34]. Second, there is the permanent yield gap that maintains this unbalance between supply and demand of food leading to a continuously low per capita food availability [35]. This is severely amplified by important food that lost up to 30% of their harvested products [33,36].

Most data show a gap between food demand and food supply in Africa and this will increase with rapid urbanization [37]. Country income level is not enough to guarantee sufficient access to food. Additionally, agronomic solutions are not genuine solutions to food security. Exploring the food-system framework gives better access to quality food and that means producing more than enough staple food [11]. Many factors affect food production such as the current climate, agricultural practices, market dynamics, existing infrastructures for food conservation and transformation, etc. Many countries that are food sufficient rely on their economic performance (cash return from other commodities) to alleviate food and nutrition gaps by enabling food purchase elsewhere [11]. In general terms, gross domestic product (GDP) is inversely correlated to the prevalence of an undernourished population (Figure 3).

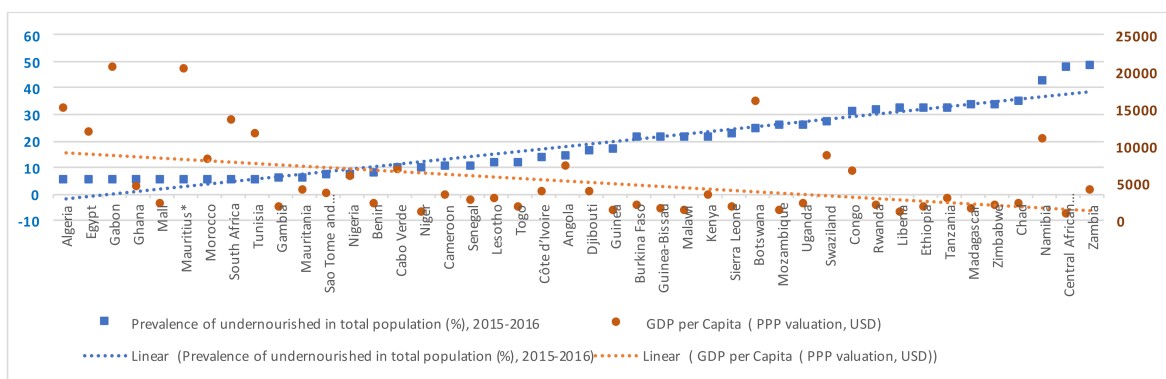

**Figure 3.** Gross domestic product (GDP) and undernourishment [28].

## 3. Places for Land Policy Improvement

### 3.1. Demography and Urbanization

Africa has the quickest population rise and urban development in the world. Africa's population is 1.2 billion and predicted to double by 2050; this represents half of the world's population growth, leading to pressure on land resources, such as agriculture, water, energy, and rapid urban development [38,39]. The continent also has the highest gains in life expectancy, rising by 6.6 years between 2000 and 2015 [40]. The land challenge of population growth, beyond demand of food, water, and energy, is that about 40% of Africans live in urban areas. This is expected to reach 56% to 60% by around 2050 [41]. Rapid urbanization not only encroaches on fertile land and natural ecosystems but two-thirds of the investment in infrastructure will occur in these built environments without compensation in natural habitat savings.

The SDGs challenges will be acute under the rapidly growing cities in Africa. Urbanization has major implications on the way land is used within the urban region and in remote rural places. The food market and demand of commodities influence so much of the land processes in rural areas. For example, it is clear that meeting urban food demand will become a major task for planners and policy makers, in particular when the urban context is dominated by an informal sector for food supply, and densely populated slums. Land stakes of urbanization will be related to a steady increase in the demand for meat, fats, fiber, and oils [37].

The rapid demand on natural resources, together with increased population and development of infrastructure, requires a strong attention to resources' limits and sustainable management of renewable and nonrenewable resources.

### 3.2. Land-Based Mitigation of Greenhouse Gas (GHG) Emissions

Carbon emissions come, to a large extent, from land use change in agriculture and forestry (>80%) [42]. Other sources include the burning of fossil fuels and the manufacture of cement. The importance of emissions here is not because of their global significance (all of Africa shares only 3.6% of global emissions, 2.3% if only Sub-Saharan Africa) [43]. Carbon emissions from agriculture, forestry, and other land use (AFOLU) are indicators of unsustainable practices for land use, such as slash and burn agriculture, and energy efficiency such as the unsustainable use of wood energy. The carbon intensity of current practices, or the amount of carbon per unit factor of activity, is still quite high and reflects the inefficient combustion of machineries and practices such as slash and burn agriculture, soil erosion, deforestation, and land degradation. Using fires for managing land is a major source of direct $CO_2$ emission [44,45] with a severe impact on the soil, biodiversity, and water cycle in Africa.

AFOLU greenhouse gas emissions in Africa account for 15% of the global total, with an annual increase of 1.6%. Livestock-related emissions from enteric fermentation and manure contribute to nearly two-thirds of the total (about 39%). Manure left on fields and fires are respectively about 28%

and 21% [46]. Most responses to GHG emissions depend on a combination of land related solutions from livestock and pasture management to agriculture and drivers of deforestation.

## 4. Opportunities for Improved Land Interventions

### 4.1. Land Restoration, Farmlands and Forestry

Several SDGs, namely SDG 2, 3, 6, 7, 8, 11, 12, 13, 14, and 15 have a strong reliance on land and improved management [47]. Land management options that contribute to the delivery of SDGs are a priority in Africa. Across all developmental sectors, SDGs cannot be met without dealing with climate change impacts on smallholder farmers, on forest, and on other land use such as rangelands and wetlands. We need to achieve these goals without affecting ecosystem services from the land. Sustainable land management becomes a pressing political challenge in Africa to help accelerate change towards achieving the SDGs. In Africa, there is large potential for scaling up various land practices to help increase productivity while reducing the environmental footprint.

Improved agricultural productivity through sustainable intensification will, for instance, reduce poverty (SDG 1) and improve nutrition for millions of poor dwellers (SDG 2) [1]. This will have a direct benefit for human health and well-being (SDG 3) [48]. Land management, including reforestation, agroforestry, restoration of wetlands, mangroves, riparian forests, and reduced erosion, will improve soil properties and mostly water recycling (SDG 6) [49]. In Africa, over 80% of the population depends on firewood and charcoal for cooking [50]. With bioenergy from the land Africa has great potential for affordable and clean energy from hydropower, solar energy, and biomass (SDG 7) [51]. Sustainable land management in and around cities can contribute to greener cities with the effect of buffering air pollution and severe climate extremes (SDG 11). The role of land in climate action is prominent in Africa where AFOLU is the major source of greenhouse gas (GHG) emissions (SDG 13). Land management based on preserving basic ecosystem functions and preserving biodiversity and ecosystem services strongly supports SDG 15 [19].

Harnessing solutions for all these SDGs, is a complex process because of the interconnected nature of the goals. Therefore, an uncoordinated implementation in places may lead to perverse outcomes. Some apparently positive actions on land could have rebound effects that are detrimental to SDGs (e.g., SDG 13 and 15 on climate and life on land, respectively). For instance, successfully providing energy access based on unsustainable wood products could come at the expense of climate goals with subsequent deforestation and forest and soil degradation [52]. One the biggest land challenges, where most of the challenges converge, is the rate of deforestation in Africa: how to conceal land productivity and the preservation of ecosystem functions [19,53]. The FAO Global Forest Resources Assessment, in particular, shows a continuous loss of forest cover at a rate >0.6% a year in West and Central Africa and >0.4% in East and Southern Africa (Figure 4).

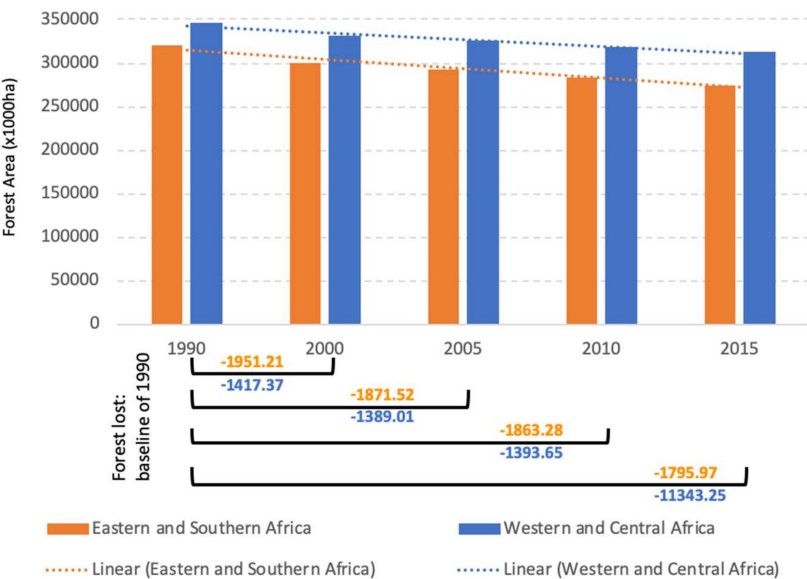

**Figure 4.** Forest area lost in Sub-Saharan Africa (SSA) [54]

Agriculture is the main driver of deforestation, but the policy framework is at the moment skewed towards large-scale use of fertilizers. The African Union (AU) Malabo Declaration on Agriculture and Postharvest Losses in 2014 sustained the pledge of the Comprehensive African Agriculture Development Program (CAADP) in 2004 for the increased use of nitrogen fertilizers to boost food production on the continent. The denial of fertilizers is not fair for countries where the yield gap is one of the big factors of food insecurity, but, it is important in establishing such policies to set safeguards and improve knowledge for optimal use. Notwithstanding the price of fertilizers that average farmers cannot afford without subsidies, it is important to question the way they are used in the African landscape. It is known that targeting the plant rather than the soil may save time, energy, and labor in applying fertilizers [55]. For instance, fertilizers are synthetic components with key nutrients such a N (nitrogen), P (phosphorus), and K (potassium). At the moment of application most crops do not need all these components. Consequently, the wastage and oversaturation of non-used components can cause many environmental problems [56]. Tools and methods related to precision agriculture can help [55].

Not in the least, it is worth mentioning that soil with very low levels of organic materials is unlikely to retain mineral fertilizers. It important therefore to combine natural solutions that incorporate organic materials such as mulching, agroforestry, manure spreading, reduce erosion, limitation of fires, cover cropping, etc. [57]. Putting these options into practice is another challenge if priorities are not vested on new economic and policy models to accelerate food security while saving land health (Table 1 shows some options of sustainable food production).

**Table 1.** Opportunities for addressing various sustainability goals through improved land management.

| Key Considerations | Selected Obstacles | Implications for Land |
|---|---|---|
| Sustainable Development Goals intend to sustain wealth and development in a responsible way; Land productivity in agriculture and forestry have to increase to meet these expectations. | Limited use of the diversity of food products in SSA; Land tenure challenges and land grabbing; Access to energy and water; Access to quality seed and seedlings; Limited capacity for food transformation and transport; Climate variability. | Sustainable intensification of agriculture; Agroforestry, land restoration, and diversification can, with appropriate land management systems, provide a range of goods, benefits, and services simultaneously, providing nutritious food, renewable energy, and clean water, while conserving biodiversity. |

**Table 1.** *Cont.*

| Key Considerations | Selected Obstacles | Implications for Land |
|---|---|---|
| Land area needed for various SDGs to accelerate current production levels are available in Africa but require appropriate management and use to reduce land degradation and restore land extensively mostly in arid and semi-arid ecosystems. | Limited investment on land restoration; Persistence of unsustainable farming and practices (e.g., soil amendment with little use of manure); Increase human and livestock pressure on natural habitats; Rapid urbanization. | Several approaches exist that increase land productivity at a low cost and with positive impacts on the environment and societies, promote efficient, multifunctional land use, increase domestication of trees with high nutrition values, and establish standards for balanced use of land in various contexts. |
| The inequity in land tenure and access to resources at the local scale leads to conflicts and contestations, and competition for land that reduces land productivity and increases inequity. | Strong land competition and conflicts related to land mostly with increasing climate change impacts; Women's access to land is still very low. | Improved land management is an institutional response to contested resource access, allowing gender and social equity enhancement and source of empowerment. |
| Development challenges and international market drivers lead to many local land challenges, but a sectoral approach that dominates govern ment systems cannot address these negative consequences in the context of SDGs. | Little support for youth enterprise development based on natural resources; No innovative investment in natural resources; Separation between water energy and agriculture sector; Lack of cohesive political frameworks. | Land restoration and combining trees with crops and livestock as an integrative approach for agriculture to help create synergy between the various SDGs in multifunctional landscapes; Break out of national silos and support resilient production systems. |

Investments in sustainably managed trees and forest products, and commodities verified and certified as 'green' as part of the Green Economy initiative contribute to the achievement of SDGs. Trees provide both food security and income through commodity market sales, and more importantly serve as a living savings account for poor people [3]. This system provides farmers with the flexibility to react to market incentives for either farm products or off-farm products. Strengthened rights and tenure through better governance of forests and trees improve local social capital—livelihoods, cultures, and aspirations—as a component of poverty alleviation.

African nations established several policy dialogues and consultations under the United Nations Economic Commission for Africa (UN-ECA) that recognize that achieving the SDGs requires an integrated approach that addresses the urgency and magnitude of the challenges. This calls for political, technical, and financial support to put in place a mechanism that ensures the a of information—most of them still missing—across sectors and scales facilitates coordination and ensures harmonization, planning, implementation, and monitoring of progress. We can learn a lot from existing science, technology, and innovation (STI) that are well integrated into national development strategies to raise productivity, improve competitiveness, support faster growth, and create green jobs [58].

*4.2. Transforming Agriculture and Food in the Context of SDGs*

The increase in food demand will be satisfied using the same land base and will have several environmental impacts [47]. Agriculture in SSA is low in all index dashboards in terms of yield, soil fertility management, balance between food and commercial crops, access to land, etc. According to estimates by The McKinzy Global Institute [56], Africa has around 600 million hectares of uncultivated arable land. This constitutes approximately 60% of all the arable land in the world. Data from African-wide surveys of land degradation by the Montpellier Panel [59], reporting on conserving, restoring, and enhancing Africa's soil, show that about 65% of the cultivated lands in Africa are infertile due to soil erosion and high population growth resulting in poor crop yields. Without significant investments in the restoration of productivity, these lands will be unviable for food production. The

consequences of land degradation and poor farming practices are that the continent cannot grow enough food to feed the growing and increasingly urbanized population.

At the same time, food security will influence the emergent properties of other drivers and pressure future policy decisions in the agriculture, forestry, energy, and conservation sectors, with different demands for land to supply multiple ecosystem services, usually intensifying competition for land in the future [60]. Growth in the demand for food in Sub-Saharan Africa (SSA) is among the highest in the world, but Africa is uniquely positioned to meet the challenge of feeding itself and the world, and spur economic growth [18]. The land potential of Africa is real yet unused [59]; the sources of food are many and a large portion of that diversity is not formally part of national food policies. The "neglected crops", including trees producing nutritious fruits, have not been formally developed despite the growing recognition that these may be the quick wins to solve instant food and nutrition demand [61].

This reference to untapped food should be articulated with the dominated smallholder agricultural model that needs to be boosted to improve productivity through better access to yield-enhancing innovations [62]. In Africa, about 30% of soils in croplands are unresponsive to fertilizer use despite being nutrient-deficient, likely due to loss of biological fertility [63]. With over 12% of the world's population, Sub-Saharan Africa accounts for less than 1% of global fertilizer demand [64]. One way to increase soil organic matter and carbon stocks is to improve tree density in farmlands, and proceed regularly with manure dispersal, mulching, or biomass offsetting, to compensate for exported biomass following harvests. Many studies report increased yield based on improved agroforestry and biomass offsetting. Yield can increase from 15% to 35% in the case of nitrogen-fixing trees in agroforestry [65].

The whole agricultural transformation can be assessed by comparing current baselines against required performances from various places or the required transformation needed (Figure 5).

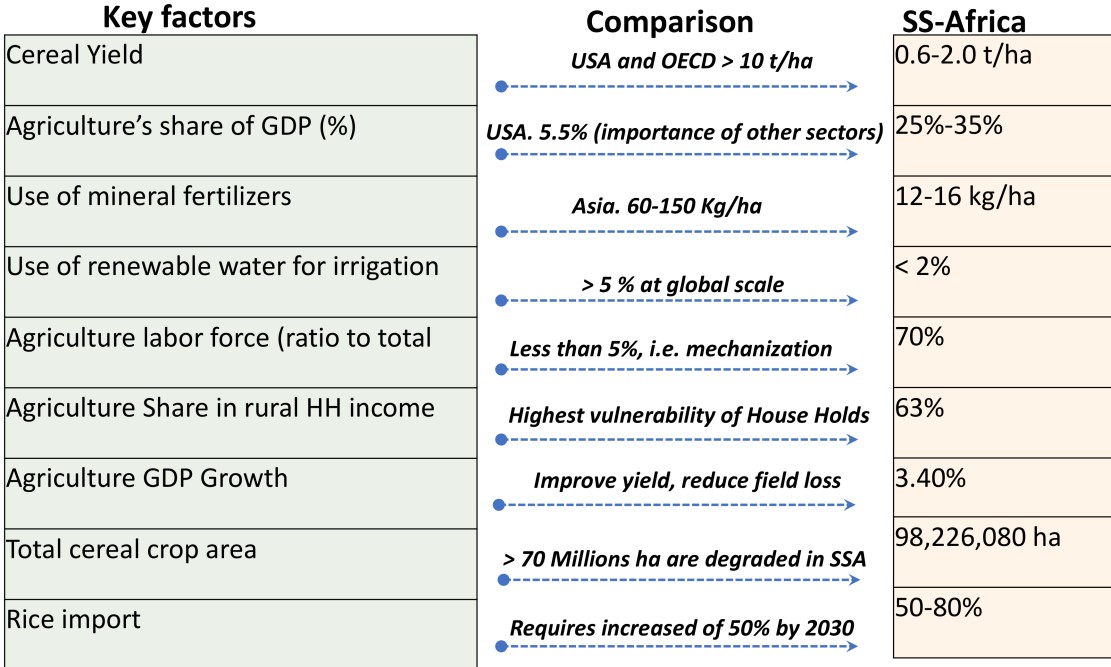

**Figure 5.** Key agriculture transformation gaps in Sub-Saharan Africa.

Technically, various production systems can become sustainable if resilient approaches are established [19]. A starting point is to assess and establish some level of tree-cover using suitable species that support agroecological function for an optimum yield, while supporting local livelihoods and maintaining soil health [66]. Larger gains can be achieved if we understand the circumstances in which productivity is significantly increased through integrated approaches of crop production, rather

than full and exclusive options for mineral fertilizers [61]. The application of an integrated approach needs to be for both small and large private agri-businesses. Supportive measures such as improved seeds, better use of fertilizers, soils management, food conservation to avoid food losses, and market incentives, require careful attention to the promotion food security. The biggest challenge, as shown in many documents, is the use of fertilizers. Better management of mineral fertilizers (i.e., precision agriculture and locally-formulated blends of nutrients), reduces overuse of the expensive products and the environmental footprint.

Moreover, for agriculture to respond to a robust greening of the economy, it is urgent to focus on land tenure access and equity, farmers' access to quality seeds and seedlings, the necessity to improve institutions, and a community's capacity building. Transitions to income-based food security such as using cash crops to access food is prevalent in many agricultural contexts and has been since colonial times. Yet nowhere have cash crops secured enough gains to secure access to food through current markets. It is therefore important to balance food and cash crops using a deliberate approach of land allocation based on market opportunities and food demand of communities.

Food production is also largely secured through urban and peri-urban agriculture [67,68]. Landscape level resource diversity and rural–urban linkages are needed for meeting food security, health, and climate change goals, beyond the requirement for sustainable cities (SDG 11). More specifically, re-thinking urban agriculture and consumers' demand for food promotes responsible and sustainable consumption, including reducing food waste and the gendered models for shared responsibility to support healthy diets to reduce growing non-transmissible diseases.

Food production is intrinsically related to energy production and use. The models promoting renewable energy compete with emerging opportunities such as increased oil exploration in Africa, leading to many new reserves found in West, Eastern, and Central Africa. Will this new venture for high-value fossil fuel sources of energy hinder the ability to explore innovation in renewable energy? The possibility of losing the possibilities offered by renewables such as solar, hydroelectricity, and biomass is likely. If full consideration is given to forest–water–energy connections, this offers a framework for a nexus approach for the sustainable use of inputs such as water and energy to accelerate food production. The role of forests goes beyond the carbon dimensions that dominate the climate change debate. First, forests, water, and energy are the foundations for carbon storage. Second, forests provide a cooling effect that buffers climate extremes and supports recycling of water and nutrients. These functions must be included in monitoring and tracking SDG indicators [69].

Water and land are likely to present the greatest challenges on the food supply side, given the declining availability of arable land (and per capita land available) and water resources in most parts of Africa. SDG 6 on access to clean water for all requires substantial improvement of watershed management. Land restoration in watershed can help improve landscapes to retain normal 'flow persistence' while providing natural processes of water recycling for quality water [70]. Many of the most significant watersheds are transboundary ecosystems that require improved regional policies to harmonize management. This calls for stronger management in watershed buffering functions through a holistic land perspective for the hydrological cycle. This includes vegetation influences on water recycling and soil moisture to deliver various ecosystem services such as those related to mangroves that are recognized as essential for both marine fish and coastal zone protection. Targeting agricultural water shortages requires improved water management, small-scale irrigation mostly during dry spells, and drip irrigation for trees and horticulture.

Regarding energy aspects, as the demand for liquid biofuels increases (especially for transport and heat combustion engines), along with food transformation and distribution needs, smallholder farmers in Africa can be linked to several value chains associated with liquid biofuels and co-products (seed oil conversion to biofuel). Careful attention should be given to the competition for land between food production and bioenergy production. Current development models could explore access to clean energy and reduce the reliance on wood fuel that is one of the major drivers of deforestation [71].

### 4.3. Climate Responsive Actions for Land-Based Development

Many of the potential GHG removal options are land-based and at the same time, options are available to reduce the risks to natural and managed ecosystems (e.g., ecosystem-based adaptation, ecosystem restoration and avoided degradation and deforestation, biodiversity management, sustainable aquaculture, and local knowledge and indigenous knowledge) [47,71,72]. The agriculture, forestry, and other land use (AFOLU) sector is responsible for over 80% of direct global anthropogenic greenhouse gas emissions in Africa [42]. Mitigation options to meet the Paris Agreement—as stated in most African nationally determined contributions (NDCs)—are significant from agriculture and forestry [73] where GHG emissions can be reduced through a range of land management practices [43,53,55,72,74].

The African land-based mitigation potential is estimated to be 265 million tons $CO_2$ per year up to 2030 through cropland management, grazing land management, and the restoration of degraded lands [75]. An additional 812 million t $CO_2$/year can be mitigated by preventing deforestation driven by agricultural expansion and through forest conservation combined with sustainable intensification practices that are capable of achieving food security [76]. Some mitigation options are described in Table 2.

**Table 2.** Feasibility action for reducing greenhouse gas (GHG) emissions.

| Options | Feasibility | Source |
|---|---|---|
| Reductions in $CH_4$ or $N_2O$ emissions (non-$CO_2$ gases) | Croplands management including paddy rice, grazing land management and livestock feed, peatlands conservation, fires control. | [10,74] |
| Conservation of existing carbon stocks | Through Reducing emissions from deforestation and forest degradation (REDD+) programs, community forestry, protected forest and conservation, urban green space, conservation of mangroves, sustainable forest management, conservation of forest biomass, peatlands and soil carbon, conservation agriculture or agroecology. | [77,78] |
| Reductions of carbon losses from biota and soils | Through management changes within the same land-use types, such as improved rotations, crops, tillage, and residue management, or by reducing losses of carbon-rich ecosystems, such as reduced deforestation and rewetting of drained peat lands, agroforestry, low tillage, and other Climate Smart Agriculture (CSA) actions. | [53,79] |
| Enhancement of carbon sequestration in soils, biota, and long-lived products | Increases in the area of carbon-rich ecosystems such as forests (afforestation and reforestation), increased carbon storage per unit area (e.g., increased stocking density in forests), carbon sequestration in soils, and wood use in construction activities. | [61,80] |
| Provision of products with low GHG emissions | Replace products with higher GHG emissions with those delivering the same service with GHG footprint (e.g., replacement of concrete and steel in buildings with wood and some bioenergy options; these options should not hamper social development where they come from). | [50,81] |
| Reductions of direct emissions | Precision agriculture, optimal use of agricultural machinery, fire control, tillage reduction. | [82,83] |
| Reductions of indirect emissions | Production of fertilizers, emissions resulting from fossil energy use in agriculture, fisheries, aquaculture and forestry or from production of inputs, though indirect emission reductions are accounted for in the energy end-use sectors (buildings, industry, energy generation, and transport). | [83] |

## 5. The Enabling Environment for Managing Trade-offs in Land Use

In Africa, many development policies are rather interventionistic and palliative. They are based on "damage and fix" approaches that are in essence rather patchy and ad-hoc. That is because of the myriad of urgencies that motive the rush of fixing burning issues rather than building a solid inclusive development process. The burden of mismanagements of resources has serious implications on addressing social impacts of economic policies that are prerequisites for resilient societies [12,56]. One of the principles of the SDG "*No-one is left behind*" could be a hypothetical concept if no effort is invested on inclusive societal structures to bring opportunities based on natural resources, particularly for the youth and women. On the land sector, the issue of land tenure equity becomes a prerequisite for land solutions to achieve SDGs [84].

The enabling environment should act as a set of policy options to support social transformation in a context where the same piece of land is used to achieved multiple objectives, including financial incomes from the trade (export) of commodities and natural resources. The SDGs are a unique opportunity to create local transformation of these commodities and influence various actors along the value chain of these products. Natural resources valuation is a place where we can demonstrate actionable opportunities, rather than just hammering problems that inhibit creativity. Innovation that leads to transformation requires several supporting land-based innovations spanning resources management, issues of access and equity, and navigating tension on land competition. For example, in land conservation and agriculture, there is a need for trade-offs to avoid compromising social demand while pursuing ecological benefits. However, across land options, there are many interventions that will be 'no-regret' options such as sustainable land management, soil fertility amendment, seeds and seedlings, markets and value chains, assessment of needs, and suitable actions [85].

There is big potential to apply the political, economic, and social incentives to empower both vulnerable communities and the private sector to create jobs and catalyze the processes of change and share benefits to a wider mass of vulnerable groups, particularly women and the youth. Adding to that, better market incentives and reducing trade barriers could be a unique opportunity to speed up wealth and spread the benefits throughout society. This requires new governance structures to embrace new power dynamics with increasing decentralization and devolution processes and a strengthening of domestic environmental knowledge. More specifically, it is important to i) ensure social protection and improve participation and equity; ii) develop synergetic cross-border policies that address sustainability; and iii) put in place a governance system that overturns the business as a usual sectoral approach to a more inclusive development pathway.

There are some co-benefits and some trade-offs associated with meeting these challenges. Addressing the competing priorities of the 17 SDGs requires explicitly addressing these aspects among stakeholders and building collaborative relationships. This provides the opportunity to achieve coherence in policies and actions across all levels and scales, from local to global. Through the SDGs Center for Africa, the African governments want to establish a support institution for an integrated approach that considers interlinkages between SDGs and addresses the urgency to achieve SDGs simultaneously. Therefore, increased coordination and harmonization in planning, cohesive implementation, and monitoring processes at the landscape, sub-national, and national levels will be serious options to face challenges [24].

A large action area will rely on managing trade-offs and synergies. For instance, conflicts may arise over the use of limited land for energy production as opposed to biodiversity conservation [86,87]. Core areas for the expansion of the global protected area network should be compared with the renewable energy potential available from land-based solar photovoltaic, wind energy, and bioenergy. A recent study [87] found that the extent of risks and opportunities deriving from renewable energy development, is highly dependent on the type of renewable source harvested, the restrictions imposed on energy harvest, and the region considered [47,87]. It appears that bioenergy production is a major potential threat to biodiversity, while the potential impact of wind and solar on land appears smaller

than that of bioenergy [5]. Africa should not use their most fertile land for bioenergy rather than producing food for a growing population.

Trade-offs can be seen between biodiversity and expansion of cropland for delivering food security. It is reported in Africa (humid tropics) that areas of high biodiversity coexist with high food insecurity communities with high risk of agricultural expansion [88]. There is an increasing risk that addressing food security through agricultural expansion could lead to biodiversity loss through damage of natural habitats [89]. In Smith [47], we found that risk of agricultural expansion overlaps significantly with areas of threatened species richness in many parts of Africa (Figure 6). This may encourage that the challenges of food insecurity and biodiversity loss are tackled together.

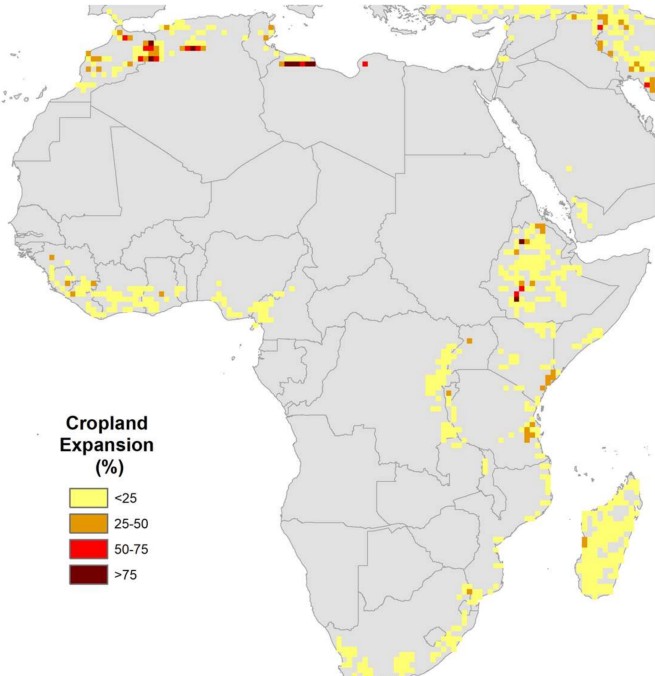

**Figure 6.** Global forecasts, under a "business as usual" scenario, of cropland expansion into biodiversity hotspots from 2010 to 2050 under Shared Socio-Economic Pathway-2 (SSP2) as predicted by the Integrated Model to Assess the global Environmental (IMAGE) model [90]. Courtesy of Amy Molotoks.

Molotoks et al. [90] found that the cropland expansion projected by 2050 is to result in a substantial loss of habitat in biodiversity hotspots such as in Ethiopia, Madagascar, the South Coast of West Africa, and South Africa, with some sites losing part of their last remaining habitats.

## 6. Conclusions and Recommendations

Land is central to the success of SDGs in Sub-Saharan Africa. Most SDGs are based on land performance indicators. Current land use approaches show divergent images. Some are very promising and already have good impacts, many others are worrisome enough to cause major caveats in the implementation of SDGs. Therefore, for the SDGs to succeed in Africa in just 10 years, the African continent should depart from business-as-usual approach and seek the means of implementation that can help accelerate changes.

The implementation of SDGs to build resilience and sustain land resource production gains will require paying simultaneous attention to the following five overarching issues:

1. Targeting places where rapid changes can facilitate positive effects and spin-offs to other development goals will be more realistic. Managing rapid urbanization and achieving land-based development while reducing GHG emissions are good options for integrated SDG policies.

2.  Closing yield gaps through sustainable intensification innovations that combine production and preservation of ecosystems' essential functions. Food security is possible in SSA where land, water, and energy are available and where labor is not a constraint. The enabling factors should be understood and addressed.

3.  Several land-based options exist to unlock natural resources' potential for people's benefits; land restoration, forest management, agroforestry, soil fertility management, agro-ecological practices, and many more are not yet up to their optimum in SSA.

4.  Identifying the trade-offs or even synergies between development goals requires simultaneous attention to several resources management to design new frameworks toward a dynamic bio-economy focus on the best practices of land resources management.

These leading orientations towards a differentiated set of land-based approaches for communities with varying internal structures and functions can be established, if the right investment and attention is given to the current challenges. Such transformative designs need to be based on the collective capacity of diverse actors to access a range of new developmental approaches. These approaches should not be based exclusively on economic options but on a careful selection of land-based innovations to accelerate development without prejudice of the environment and social benefits.

This transformation will happen during a period of global geopolitical turbulence with a growing threat of economic protectionism and environmental turbulence with extreme weather events bringing several lasting damages to Africa. For instance, recent drought in SSA has ignited many conflicts with consequences of malnutrition and lack of food security. It is important therefore to bring new revolutionary ideas on how to pioneer/incubate new approaches to address complex, urgent, and long-term changes using novel approaches that cut across disciplines, social statuses, issues, and organizations, for example, to produce enough food, transform most of it locally, and adjust the markets to favor wide access to food products.

The challenges of such transformation are how to influence social structures to adopt new and improved land management strategies leading to human well-being and equitable rights, transparent governance, and justice. Sub-Saharan Africa has to transcend the institutional barriers (most governments still apply the sectoral approaches for development), and patterns of stratification related to diverse epistemic groups with chattered development goals. Social change in particular, has always involved oppositions of ideas because of the diversity of interests between different groups, even within a single country, and requires the redistribution of resources and institutional improvements for new forms of governance that manage collective concerns at different levels.

Moreover, it is important to highlight that all land-based innovations may not be appropriate for sustainable transformation, therefore, working on adequate co-design approaches is key to addressing urgent challenges. New orientations for rapid transformation will start and stay consolidated by establishing new values and revisiting the narratives on SDGs that are relevant to African specificities. All innovations may not be appropriate for sustainable transformation, therefore, working on linking adequate approaches with suitable contexts using co-design, governance, and where applicable, a bottom-up-approach are key to addressing these urgent challenges.

**Author Contributions:** All authors have read and agree to the published version of the manuscript. Conceptualization, methodology, formal analysis, draft preparation, writing by the author C.M.

**Funding:** This research received no external funding.

**Acknowledgments:** The author thanks the ICSS (International Conference on Sustainability Science), held on 21–23 August 2017 in Stockholm Sweden; and the Seedbeds of Transformation-The Role of Science with Society and the SDGs in Africa (9–11 May 2018, Port Elizabeth, South Africa), where the initial ideas were presented. A big thank to the reviewers for their valuable inputs.

**Conflicts of Interest:** The authors declare no conflicts of interest.

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
