# Peer review of "Use It Sustainably or Lose It! The Land Stakes in SDGs for Sub-Saharan Africa"

_land, doi:10.3390/land9030063_

Round 1

Reviewer 1 Report

The topic is of very high interest for African policymakers. And for this reason it should be improved in a way that make it more easily readible for them.

More specifically, 3.2, 4, and even 5, should include graphics representing all the different mechanisms and strategies proposed in the text (maybe using the different SDGs and the different strategies proposed for each SDG). The conclusions/recommendations are very long and maybe they could be splited in different bullet points to make them clearer for the reader.

Figure 5 should be improved and reviewed. Yields are in t/ha or Kg/ha?

I miss one important point in the manuscript. What about "land grabbing"? It is not explicitly mentioned in the paper, but it is getting more importance since the last years.

In summary, I strongly suggest adding some graphics in order to make the manuscript clearer for the reader.

Author Response

All responses are in the file upload below.

Reviewer 2 Report

I suggest you rethink the message you want to get across and restructure your article in an ordered manner

Author Response

All responses in the file uploaded below.

Reviewer 3 Report

This paper addresses a theoretical analysis how Africa could develop and consolidate winning land practices to address poverty, growing population and rapid urbanization. The analysis relates several important aspects as the food demand, demography, environmental issues as carbon emissions, geopolitical abs economic aspects, policy framework, etc. The study is based in a literature and data review, but without statistical analysis relating foreseeing different scenarios. The conclusions are based in personal interpretations. Nevertheless, the paper is interesting, so I recommend their publication.

The abstract should include the objective of the manuscript.

Author Response

All responses in the file uploaded below

Reviewer 4 Report

The article is very interesting and useful because it provides a large-scale framework aimed at implementing new policies at national or regional level. The structure of the work tries to provide some useful frameworks for this purpose.

Some indications and reference frames are provided through tables and figures which however can be improved. In particular:

Table 1, it is suggested to add a third column in which the framework of obstacles with respect to the framework of objectives is provided;

Figure 1 is unclear, data, indicators, and units of measurement should be better specified;

Table 2 reports the sources of the scientific literature for the proposed solutions, but it would be more useful to report examples and successful practices with the relative quantifications;

the range of solutions proposed (table 2) is, compared to the mentioned literature, a small enough quantity, can it be enlarged? can more works be mentioned referring to the areas of analysis and not to other regions (Mediterranean, Asia ..)?

Author Response

(The authors gave the same response as above.)

Round 2

Reviewer 2 Report

The author clearly improved on the structure and content of the paper and addressed most of my previous comments. There could still be a stronger link between conclusions and content. The conclusions are very valuable and it would be good to track back from the major points in the conclusions to the text, to see whether these points (in particular 1-5 in lines 687-700) really can be derived from the results presented. I think it is there in general, but editing could be improved so that this relation becomes more clear. In Table 1 I would move the added column on obstacles forward, to become the second column, since I assume that the opportunities address these obstacles. If that is not the case, you need to rethink the table. What do you want to present and how does this support your text? 

I see the structure now as follows:

introduction poverty and food security land restoration, agriculture and forestry (check numbering of this and following sections) transforming the agriculture-food (sector?, nexus?) in the context of SDGs achieving SDGs and land based mitigation the enabling environment for managing trade-offs in land use conclusions and recommendations

It would be good to explain in the introduction the relation between these sections and how you think these will lead to your conclusions

Author Response

The response to reviewer 2 is attached

Reviewer 4 Report

I believe two questions I have raised have remained unanswered. In particular:

1. Table 2 reports the sources of the scientific literature for the proposed solutions, but it would be more useful to report examples and successful practices with the relative quantifications; 

2.  the range of solutions proposed (table 2) is, compared to the mentioned literature, a small enough quantity, can it be enlarged? can more works be mentioned referring to the areas of analysis and not to other regions (Mediterranean, Asia ..)? 

Maybe compared to the second one I was unclear. I also believe that the literature cited refers to areas (Asia, Mediterranean, UK) that are very different from that of analysis. My suggestion is to cite solutions referring to Sub-Sahara Africa.

Author Response

response to reviewer 4 attached
